# Human substance P receptor binding mode of the antagonist drug aprepitant by NMR and crystallography

Shuanghong Chen[1,2,3], Mengjie Lu[1,2,3], Dongsheng Liu[4], Lingyun Yang[4], Cuiying Yi[1,2], Limin Ma[1,2], Hui Zhang[1,2,3], Qing Liu [2,5], Thomas M. Frimurer [6], Ming-Wei Wang [2,3,5,7,8], Thue W. Schwartz[6], Raymond C. Stevens [4,8], Beili Wu [2,3,8,9], Kurt Wüthrich[4,8,10] & Qiang Zhao[1,2,3,9]

Neurokinin 1 receptor (NK1R) has key regulating functions in the central and peripheral nervous systems, and NK1R antagonists such as aprepitant have been approved for treating chemotherapy-induced nausea and vomiting. However, the lack of data on NK1R structure and biochemistry has limited further drug development targeting this receptor. Here, we combine NMR spectroscopy and X-ray crystallography to provide dynamic and static characterisation of the binding mode of aprepitant in complexes with human NK1R variants. [19]F-NMR showed a slow off-rate in the binding site, where aprepitant occupies multiple substates that exchange with frequencies in the millisecond range. The environment of the bound ligand is affected by the amino acid in position 2.50, which plays a key role in ligand binding and receptor signaling in class A GPCRs. Crystal structures now reveal how receptor signaling relates to the conformation of the conserved NP[7.50]xxY motif in transmembrane helix VII.

---

[1] State Key Laboratory of Drug Research, Shanghai Institute of Materia Medica, Chinese Academy of Sciences, 555 Zuchongzhi Road, Pudong, Shanghai 201203, China. [2] CAS Key Laboratory of Receptor Research, Shanghai Institute of Materia Medica, Chinese Academy of Sciences, Shanghai 201203, China. [3] University of Chinese Academy of Sciences, No. 19A Yuquan Road, Beijing 100049, China. [4] iHuman Institute, Shanghai Tech University, 393 Hua Xia Zhong Road, Shanghai 201210, China. [5] The National Center for Drug Screening, Shanghai Institute of Materia Medica, Chinese Academy of Sciences, 189 Guo Shou Jing Road, Pudong, Shanghai 201203, China. [6] Novo Nordisk Foundation Center for Basic Metabolic Research, University of Copenhagen, Blegdamsvej 3b, Copenhagen 2200, Denmark. [7] School of Pharmacy, Fudan University, 826 Zhangheng Road, Shanghai 201203, China. [8] School of Life Science and Technology, ShanghaiTech University, 393 Hua Xia Zhong Road, Pudong, Shanghai 201210, China. [9] CAS Center for Excellence in Biomacromolecules, Chinese Academy of Sciences, Beijing 100101, China. [10] Department of Integrative Structural and Computational Biology, The Scripps Research Institute, 10550 North Torrey Pines Road, La Jolla, CA 92037, USA. These authors contributed equally: Shuanghong Chen, Mengjie Lu, Dongsheng Liu. Correspondence and requests for materials should be addressed to B.W. (email: beiliwu@simm.ac.cn) or to K.Wüt. (email: wuthrich@scripps.edu) or to Q.Z. (email: zhaoq@simm.ac.cn)

Substance P (SP) was the first identified mammalian neuropeptide, which was discovered in 1931 by Von Euler and Gaddum as a vasodilator substance in crude tissue extracts from equine brain and intestine[1]. The acid alcohol extracted powder was at the time referred to as SP (P for powder) and eventually identified as an undecapeptide in 1971[2]. The receptor of SP, which was later named neurokinin 1 receptor (NK1R) or tachykinin 1 receptor (TACR1), is widely distributed in the central and peripheral nervous systems[3], and is critically involved in pain[4], depression[5], inflammatory and immune responses[6], neurodegenerative diseases[3], cancer[7] and emesis[8]. NK1R evokes the release of many neurotransmitters such as acetylcholine, GABA, catecholamine, and histamine[9]. Therefore, this receptor has long been considered as an attractive drug target for the treatment of pain, addiction, anxiety, and related disorders.

Aprepitant (2-(R)-(1-(R)-3,5-Bis(trifluoromethyl)phenylethoxy)-3-(S)-(4-fluoro)phenyl-4-(3-oxo-1,2,4-triazol-5-yl)methylmorpholine; also known as MK-869 and L-754030; Merck & Co., West Point, Pennsylvania), is a highly selective NK1R antagonist. It is an FDA-approved drug (brand name: Emend) for the treatment of chemotherapy-induced nausea and vomiting (CINV)[10,11], and several related analogs have undergone clinical trials for depression[12]. Although these trials failed, potentially due to low receptor occupancy, both preclinical data and positive clinical evidence suggest that NK1R antagonists, including aprepitant, have a very distinct therapeutic action with only mild and tolerable side-effects when compared with all other antidepressants[5,12,13]. However, long after its approval in 2003, the binding mechanism of aprepitant to NK1R remains elusive due to a lack of structural information and poor understanding of the receptor biology, limiting the development of improved NK1R antagonists.

It was reported that mutations of NK1R at residue 2.50 (residue numbering using Ballesteros–Weinstein nomenclature[14]), which in class A G protein-coupled receptors (GPCRs) is highly conserved as $D^{2.50}$ or $E^{2.50}$, greatly affects its agonist binding, activation and downstream signaling[15–17]. These studies showed that upon binding to SP, the wild-type NK1R efficiently activates the Gs, Gq, and β-arrestin pathways. However, it has been reported that mutating the conserved $E^{2.50}$ to aspartic acid in NK1R reduces the Gs and β-arrestin signaling with Gq signaling unaffected[17]. Mutating this residue to asparagine in other GPCRs also exhibited diminished signal transduction[18,19]. Overall, the residue at position 2.50 of class A GPCRs is widely believed to play a crucial role in GPCR activation[20]. It has also been postulated that an extended hydrogen-bonding network between the conserved residues in the 7-transmembrane (7TM) helical bundle constitutes an allosteric interface essential for stabilizing different active and inactive conformations[17].

To provide static characterization of cognate ligand recognition by NK1R and modulation of ligand binding by the residue in position 2.50, we determined the crystal structures of two human NK1R variants, with aspartic acid or asparagine at the 2.50 position, bound to the antagonist aprepitant. For understanding the dynamic component of aprepitant binding, we conducted nuclear magnetic resonance (NMR) spectroscopy of NK1R and several functionally characterized variants.

## Results

**Overall architecture of NK1R.** To improve receptor stability and facilitate crystallization, three mutations, $Y121^{3.41}W$, $Q165^{4.60}A$, and $T222^{5.64}R$, were introduced and 10 residues (residues 227–236 of NK1R) of the third intracellular loop (ICL3) were replaced with a modified T4 lysozyme (mini-T4L). The optimized NK1R protein was co-crystallized with aprepitant, but the crystals diffracted to only about 6 Å. The resolution was improved to 3.2

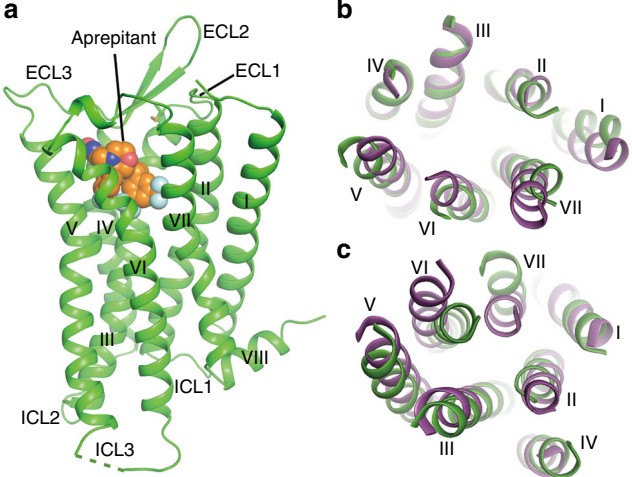

**Fig. 1** Overall structure of the NK1R–aprepitant complex. **a** Structure of the NK1R–aprepitant complex. The NK1R structure is shown as green cartoon. Aprepitant is shown as spheres with orange carbons. The disulfide bond is displayed as yellow sticks. The missing portion of ICL3 is indicated by a green dashed line. **b, c** Structural comparison between NK1R and NTSR1 (PDB accession code: 4GRV). The helical bundles of the receptors are colored green (NK1R) and purple (NTSR1). **b** Extracellular view. **c** Intracellular view

Å by replacing residue $E78^{2.50}$ with aspartic acid. To further improve the crystal quality, the residue at the 2.50 position was mutated to asparagine and $Q165^{4.60}A$ was reinstated, whereas $Y121^{3.41}W$ and $T222^{5.64}R$ were maintained. The resulting NK1R–aprepitant complex structure was determined at 2.7 Å resolution (Supplementary Table 1). The two structures are similar, with an overall Cα root mean square deviation (r.m.s.d.) of 0.4 Å. The higher resolution structure was used in the discussion below.

The NK1R structure consists of a canonical 7TM helical bundle with three extracellular loops (ECL1–3), three intracellular loops (ICL1–3), and an amphipathic helix VIII (Fig. 1a). ECL2 forms a β-hairpin structure with a conventional disulfide bond between ECL2 and helix III, a feature also observed in other solved structures of peptide GPCRs. Compared with the previously determined crystal structure of the closely related neurotensin receptor (NTSR1)[21], the extracellular tips of helices I, V, VI and VII move away from the central axis of the helix bundle by 3–5 Å (Fig. 1b). These conformational differences on the extracellular side are further transferred to the intracellular side of the helical bundle (Fig. 1c). The intracellular tip of helix VI in the NK1R structure moves inwards by 7 Å, while the tip of helix VII moves outwards by 6 Å compared to the NTSR1 structure, which adopts an active conformation. Since these structural differences were not likely induced by the fusion partner differences between the two structures[22], our data suggest that the NK1R–aprepitant structure is in an inactive state.

**Ligand-binding pocket of aprepitant.** Aprepitant is a derivative of morpholine with bis-trifluomethyl-phenylethoxy, 4-fluorophenyl, and 3-oxo-triazol groups at its 2, 3, and 4 positions, respectively[10]. It occupies a binding site in NK1R similar to that of suvorexant in the orexin 2 receptor ($OX_2R$) structure[23], adopting an extended conformation with the bis-trifluomethyl-phenylethoxy group at the bottom of the binding pocket, the morpholine and 4-fluoro-phenyl groups in the middle, and the 3-oxo-triazol group close to the extracellular surface (Fig. 2 and Supplementary Figure 1a, b).

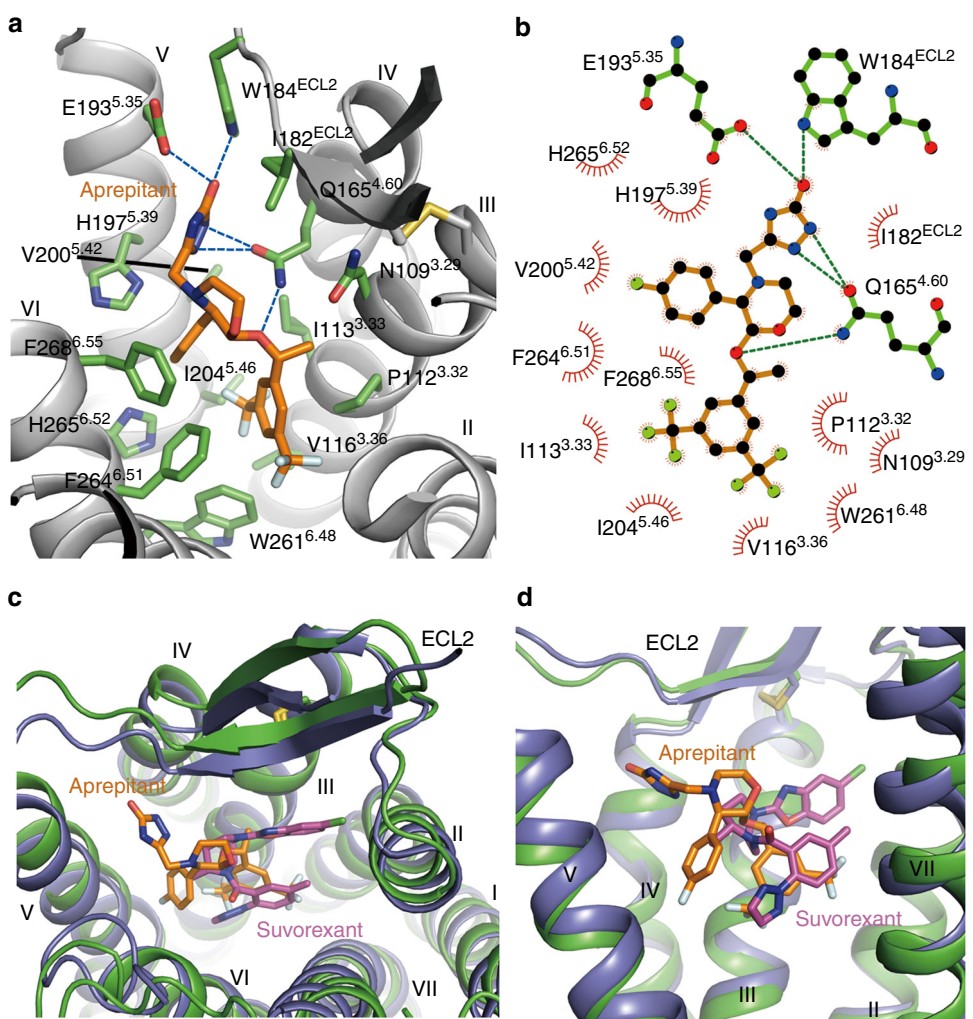

**Fig. 2** The NK1R binding pocket for aprepitant. **a** Key residues of NK1R for aprepitant binding. The receptor is shown as gray cartoon. Aprepitant (orange carbons) and receptor residues (green carbons) involved in ligand binding are shown as sticks. Other elements are colored as follows: oxygen, red; nitrogen, dark blue; fluorine, cyan. **b** Schematic representation of interactions between NK1R and aprepitant analyzed using the LigPlot+ program[42]. Polar interactions are shown as dashed lines. **c**, **d** Comparison of ligand-binding sites between NK1R (green) and $OX_2R$ (PDB accession code: 4S0V; purple). The ligands aprepitant and suvorexant are shown as orange and magenta sticks, respectively. **c** Extracellular view. **d** Side view

Multiple hydrogen bonds as well as hydrophobic interactions between NK1R and aprepitant are formed to ensure both ligand selectivity and binding affinity. The bis-trifluomethyl-phenyl is a common chemical group shared by many different NK1R antagonists. In the NK1R–aprepitant complex structure, it inserts into a hydrophobic subpocket formed by helices III, V, and VI, and serves as an anchor to stabilize the morpholine group in an optimal orientation (Fig. 2a). Together with the benzene ring, the two trifluomethyl groups pinch W261[6.48] and inhibit the activation of NK1R by preventing the toggle switch[24] of this residue. F264[6.51] and P112[3.32] form extensive interactions with both trifluomethyl groups as well as edge-π interactions with the benzene ring on either side. Additionally, the trifluomethyl groups make hydrophobic contacts with I113[3.33], V116[3.36], and I204[5.46]. These interactions serve to fix aprepitant in the binding pocket while mutations of the key residues greatly weaken the antagonism activity of aprepitant (Supplementary Table 2).

The bis(trifluomethyl)phenylethoxy-morpholine-(4-fluoro) phenyl group adopts a horseshoe shape conformation with the (4-fluoro)phenyl group extending into a subpocket shaped by helices V and VI in the NK1R structure, and the triazolinone group makes strong interactions with helices IV and V and ECL2

(Fig. 2a). The nitrogen atoms in the triazole group form two hydrogen bonds with Q165[4.60], while the oxygen in the triazolinone group establishes another two hydrogen bonds with W184[ECL2] and E193[5.35]. These hydrogen bonds greatly contribute to the binding of aprepitant to the receptor, as shown in Supplementary Table 2 and supported by previous studies where removing the triazolinone substituent decreased ligand-binding affinity by 30-fold[10,25].

**Structural and signaling differences between 2.50 mutants.** The overall backbones of NK1R in the two crystal structures are very similar ($C_α$ r.m.s.d. within the whole receptor is 0.4 Å). However, relatively large conformational differences of residue side chains were observed around the 2.50 residue (Supplementary Figure 1c, d). It has been reported that some conserved residues around this region form a hydrogen-bond network to regulate receptor activation[17]. In the NK1R structure with the E78[2.50]D mutation, residue D78[2.50] forms a hydrogen bond with N301[7.49] (2.8 Å). In contrast, in the structure with the E78[2.50]N mutant, the side chain of N78[2.50] rotates by ~30°, eliminating the interaction between N78[2.50] and N301[7.49] (3.5 Å) (Fig. 3a). Without the hydrogen

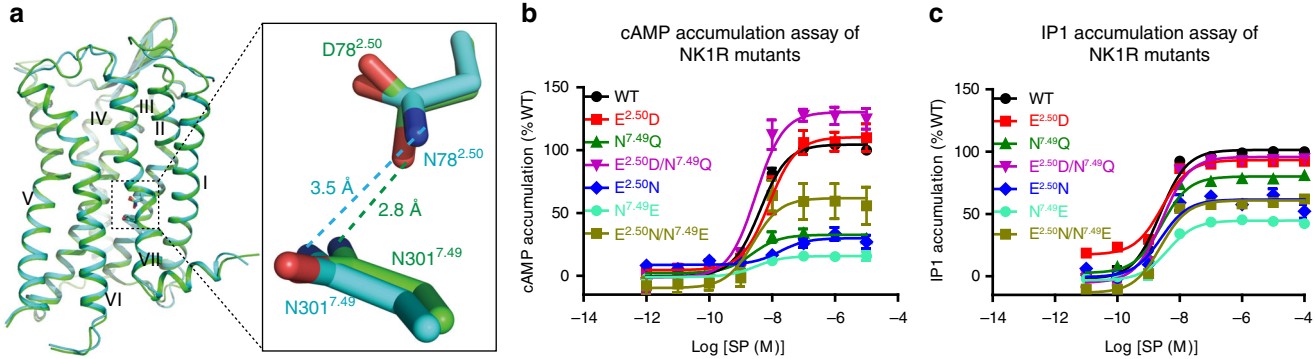

**Fig. 3** Interaction modes of the residues at positions 2.50 and 7.49 and functional assays of NK1R mutants. **a** Strong hydrogen-bond interaction between D78[2.50] and N301[7.49] (2.8 Å) and weak interaction between N78[2.50] and N301[7.49] (3.5 Å). Two NK1R structures are shown in cartoon representation and colored green (E78[2.50]D mutant) and cyan (E78[2.50]N mutant). The residues at positions 2.50 and 7.49 are shown as sticks. **b** SP-induced cAMP accumulation measurements of the wild-type (WT) NK1R and the mutants E78[2.50]D, N301[7.49]Q, E78[2.50]D/N301[7.49]Q, E78[2.50]N, N301[7.49]E, and E78[2.50]N/N301[7.49]E. Dose–response curves were generated from at least three independent experiments performed in triplicate. Data shown are mean ± s.e.m. See Supplementary Table 3 for detailed statistical evaluation. **c** SP-induced IP1 accumulation of the WT NK1R and the mutants E78[2.50]D, N301[7.49]Q, E78[2.50]D/N301[7.49]Q, E78[2.50]N, N301[7.49]E, and E78[2.50]N/N301[7.49]E. Dose–response curves were generated from at least three independent experiments performed in triplicate or duplicate. Data shown are mean ± s.e.m. See Supplementary Table 2 for detailed statistical evaluation. Source data for Fig. 3b, c are provided as a Source Data file

bond, the main-chain backbone of the NP[7.50]xxY motif moves away by approximately 0.5 Å in the N78[2.50] structure and the position of residue N301[7.49] rotates away by about 0.3 Å. This conformational change may influence receptor signaling by disrupting interactions required for activation[26]. In contrast, the interaction between D/N78[2.50] and S119[3.39], which is thought to be important for the allosteric modulation of many class A GPCRs[17,27], is relatively weak in the two NK1R structures (3.8 Å for the N78[2.50] and 3.3 Å for the D78[2.50] structures), suggesting that S119[3.39] may play a less critical role in NK1R activation. Compared to another recently solved NK1R structure with glutamic acid at 2.50 position[28], the side chain of E[2.50] further extends toward the N301[7.49] and further away from S119[3.39], in agreement with our speculation.

NK1R signals efficiently through $G_q$, $G_s$, and β-arrestin when stimulated by SP[29,30]. It has been demonstrated that mutation of the 2.50 residue greatly affects NK1R signal transduction[17]. Our results of cell signaling assay indicate that mutations E78[2.50]D and E78[2.50]N do not influence basal activity of NK1R (Supplementary Figure 2 and Supplementary Figure 3). To further understand the role of the hydrogen-bond interaction between the residues at positions 2.50 and 7.49 in receptor activation, we performed cAMP and inositol phosphate (IP) accumulation assays for the NK1R mutants E78[2.50]D, E78[2.50]N, N301[7.49]Q, N301[7.49]E, E78[2.50]D/N301[7.49]Q, and E78[2.50]N/N301[7.49]E without any fusion partner, as it would block the G protein binding and the receptor could not be activated (Fig. 3b, c, Supplementary Figure 2, Supplementary Figure 3, Supplementary Table 2 and Supplementary Table 3). Our data show that mutant E78[2.50]D retained the ability of stimulating SP-induced $G_s$ and $G_q$ signaling as seen with the wild-type receptor, while mutant N301[7.49]Q exhibited reduced $G_s$ and $G_q$ responses to SP. In contrast to the two single-residue mutations, the double mutation E78[2.50]D/N301[7.49]Q, which may benefit from the recovery of the hydrogen-bond interaction, preserved both $G_s$- and $G_q$-mediated signaling. The differences in the G protein signaling could be due to either direct influences of different mutants or lower surface expression caused by the disruption of ground state interactions of these mutants, which in turn alters the observed signaling. These results indicate that hydrogen-bond interactions between helices II and VII are required for receptor activation. Similarly, the other single mutations, E78[2.50]N and

N301[7.49]E, caused severe loss of the SP-induced $G_s$ and $G_q$ signaling, while the double mutant E78[2.50]N/N301[7.49]E restored downstream cAMP and IP accumulation levels (to 50–80%) when compared to the wild-type receptor.

**[19]F-NMR studies of aprepitant bound to NK1R.** To investigate the dynamics of NK1R and characterize the binding mode of aprepitant in solution, we acquired [19]F-NMR spectra of the ligand in the free-state as well as bound to NK1R variants with different mutations at the 2.50 position. In the n-dodecyl-β-D-maltopyranoside (DDM)/cholesteryl hemisuccinate (CHS) detergent buffer used to solubilize NK1R, aprepitant showed two fluorine signals at −62.6 ppm (M) and −113.1 ppm (A), corresponding to the two trifluoromethyl groups and the single aromatic [19]F atom, respectively (Fig. 4e, j and Supplementary Table 4). For aprepitant bound to NK1R, the trifluoromethyl resonance was split into two peaks, P1 centered at −61.6 ppm, and P2 at −63.0 ppm, both showing further fine structures (Fig. 4a–d). P1 and P2 correspond to the two trifluoromethyl groups, and the chemical shifts reflect the different microenvironments in the NK1R binding sites (Fig. 2). Lorentzian deconvolution shows that P1 and P2 each contain two components in all four complexes, and that a peak M from aprepitant bound to excess micelles overlaps with P2 (Supplementary Figure 4). In all four complexes, there are thus two different substates of the bound aprepitant (Fig. 4). Based on ring current shift calculations using the crystal structure[31,32], we assigned the peak P1 to the fluorine atoms FAD, FAE, and FAF, and peak P2 to FAI, FAG, and FAH (Supplementary Figure 5 and Supplementary Table 4). Major contributions to the ring current shifts came from the NK1R residues W261[6.48], F264[6.51] and F268[6.55], demonstrating that the ligand-receptor interactions in the crystal structure and in solution are closely related. Variable chemical shift dispersion between P1 and P2 in the different mutants (Fig. 4a–d) shows that there are conformational differences in the binding site. When the mutation E78[2.50]D was introduced, the fluorine peaks P2a and P2b (−63.1 and −62.9 ppm, respectively) remained separated. For the E78[2.50]N mutant, the two peaks P2a and P2b merged accidentally into a single peak, while the peaks P1a and P1b remained separated. These data suggest that the ligand-binding pose preference is critically dependent on the

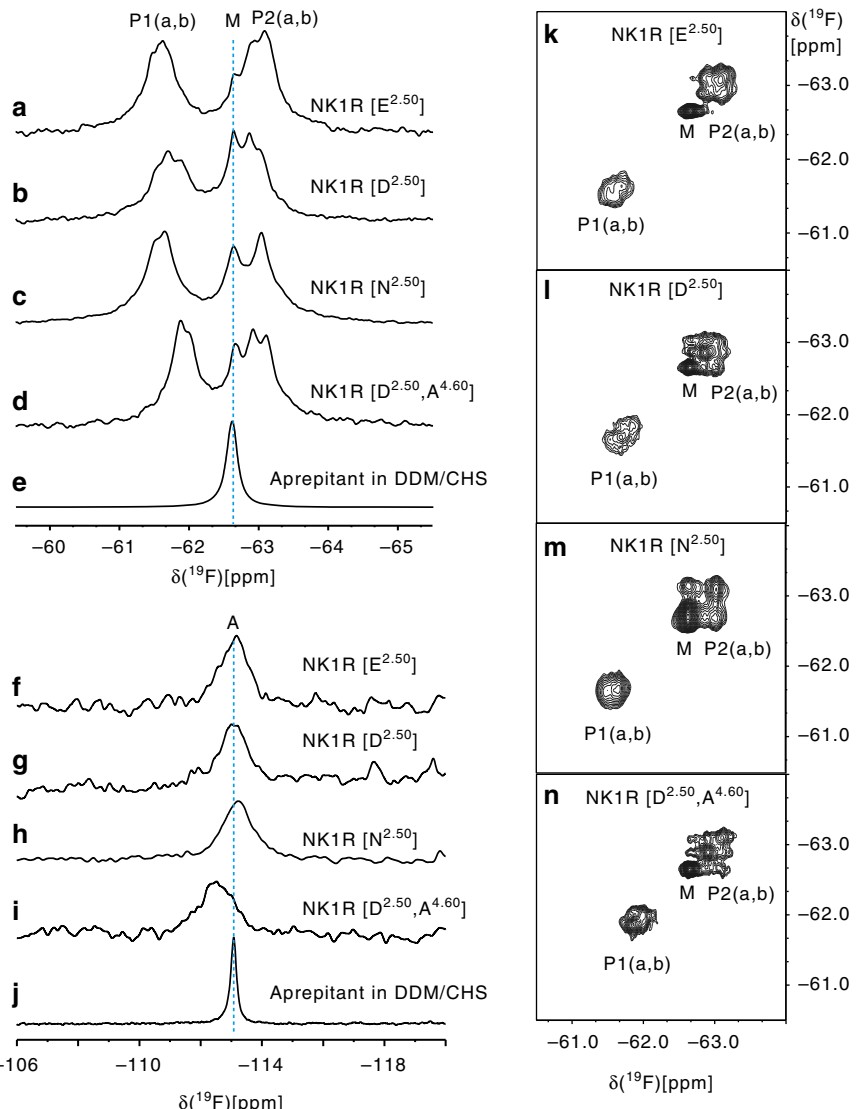

**Fig. 4** [19]F NMR studies of aprepitant bound to NK1R in solution. **a–j** 1D [19]F-NMR spectra of aprepitant in complex with different NK1R mutants and aprepitant in the DDM/CHS detergent buffer. **a–e** Spectral region showing the resonances of the trifluoromethyl groups of the bis-trifluoromethyl-phenylethoxy moiety; **f–j** Spectral region of the 4-fluorophenyl moiety. **k–n**, 2D [[19]F,[19]F]-EXSY spectra recorded with a mixing time of 150 ms covering the same spectral region as panels a–d

residue in position 2.50. This correlates with our cAMP and IP1 accumulation assay results, which show that the E78[2.50]N replacement has a larger impact on receptor signaling than the E78[2.50]D mutation.

The presence of the separate peaks P1a, P1b, P2a, and P2b implies that the corresponding conformational substates exchange slowly on the chemical shift time scale[31]. To explore possible differences among the conformational exchange rates in the different proteins, we subsequently conducted two-dimensional (2D) [[19]F, [19]F]-EXSY experiments (Fig. 4k–n). The presence of cross peaks between the components a and b of P2 shows that the exchange between these two conformational substates in the binding site has a rate in the millisecond range. This is clearly displayed in Fig. 4k–n, whereas the separation of the peaks a and b in P1 is too small to enable a detailed analysis. The absence of cross peaks between P1 or P2 and M indicates that the exchange between free and receptor-bound aprepitant is too slow to be seen with current measurements, with $k_{ex} \leq 10\,\text{s}^{-1}$ or possibly orders of magnitude slower. The absence of cross peaks between P1 and P2 shows that under the present experimental

conditions, the bis-trifluomethyl-phenylethoxy moiety did not undergo ring flipping motions[31], suggesting that the ligand-binding site environment imposes a high-energy barrier to such mobility.

## Discussion

The crystal structure of NK1R bound to aprepitant provides a detailed static picture of how the receptor interacts with its antagonist. Overall, 15 residues are involved in the binding of aprepitant, of which 10 are highly conserved across the neuro-kinin receptor subfamily (Supplementary Figure 6). However, the key residue F264[6.51], which, to our knowledge, is not known to be involved in the ligand binding of NK1R, is presented as tyr-osine in NK2R and NK3R[33]. In the NK1R structure, the residue F264[6.51] forms a strong edge–π interaction with the bis-trifluomethyl-phenyl ring of aprepitant, and the substitution of the phenyl group with the phenolic group may disrupt the interaction and cause a spatial clash with the ligand. Indeed, aprepitant and its analogs possess a 1000-fold higher selectivity

towards NK1R over NK2R and NK3R despite their highly conserved ligand-binding pockets[10,25]. When F264[6.51]Y was introduced, the inhibition in $G_q$ signaling of NK1R by aprepitant was significantly decreased (Supplementary Table 2). Besides F264[6.51], E193[5.35], and I204[5.46], which form a hydrogen bond or hydropobic interactions with aprepitant, are also not conserved among the neurokinin receptors (Supplementary Figure 4). This residue may contribute to ligand selectivity within the neurokinin receptor family. Thus, insights gained from the NK1R–aprepitant complex structures will greatly facilitate the development of more selective drug leads by targeting the variable regions of the ligand-binding pocket and enhancing interactions with corresponding residues.

Conserved residues at the allosteric interface between positions 2.50, 3.39, and 7.49 form a complex water hydrogen-bond network, thereby fine-tuning the 7TM conformation of GPCRs[17]. This network is gated by two conserved residues: W[6.48] in the CWxP[6.50] motif of helix VI within the ligand-binding pocket and Y[7.53] in the NP[7.50]xxY motif of helix VII on the intracellular side of the receptor. Comparing the NK1R structures with different mutations at position 2.50, we observed structural differences in the helical bundle around W[6.48] and Y[7.53] region, especially in the NPxxY motif, suggesting that the conserved residue E[2.50] may regulate receptor signaling via the hydrogen-bond network of this region. In addition, a dynamic perspective from NMR data demonstrates that aprepitant displays slightly different binding modes between the two NK1R structures, suggesting that ligand could adopt multiple poses in NK1R, which might be further regulated by this hydrogen-bond network. The previously reported fact that alanine substitutions of residues, such as E[2.50] and N[7.49], increased the constitutive activity $G_s$ but not $G_q$ signaling[17] is in line with our NMR results showing that only certain receptor conformations were affected by mutations of these residues. Our results imply that even though different G protein subtypes share similar structure scaffold, their activation requires a different receptor conformation that is regulated by this hydrogen-bond network. Previous research has shown that the residue at position 2.50 is very important for GPCR signaling[15–17] and that its mutation to asparagine eliminates the downstream signals on several GPCRs[18,19]. Our observations of conformational changes in the NK1R structures, which are consistent with these reports, together indicate that this hydrogen-bond network may play a key role in receptor activation and is valuable for deepening the understanding of the drug action.

## Methods

**Cloning of the NK1R receptor.** The codon-optimized human NK1R gene (sequence is shown in Supplementary Table 5) was cloned into a modified pFastBac1 vector (Invitrogen) containing an expression cassette with a haemagglutinin signal sequence and a Flag tag at the N terminus and a PreScission protease site followed by a 10× His-tag at the C terminus. To facilitate crystallization, a modified T4 lysozyme (mT4L)[34] protein was inserted between the residues S226 and H237 in the third intracellular loop (ICL3) of NK1R and 72 residues (336–407) were removed at the C terminus (primer sequences are shown in Supplementary Table 5). The NK1R-mT4L gene was further modified by introducing four mutations based on literature:[34] E78[2.50]D, Q165[4.60]A, Y121[3.41]W, and T222[5.64]R (construct NK1R-E78[2.50]D) to improve the protein yield by over twofolds and the protein melting temperature by ~10°. Another NK1R construct NK1R-E78[2.50]N was generated by switching the mutation E78[2.50]D to E78[2.50]N and removing the Q165[4.60]A mutation.

**Expression and purification.** High-titer recombinant baculovirus (>$10^8$ viral particles per ml) was obtained using the Bac-to-Bac Baculovirus Expression System (Invitrogen). *Spodoptera frugiperda* (*Sf*9) cells (Invitrogen) at a cell density of 2–3 × $10^6$ cells ml$^{-1}$ were infected with the virus at a multiplicity of infection of 5. Cells were routinely tested for mycoplasma contamination. Cells were harvested by centrifugation 48 h post-infection and stored at −80 °C until use. Insect cell membranes were disrupted by thawing frozen cell pellets in a hypotonic buffer containing 10 mM HEPES, pH 7.5, 10 mM $MgCl_2$, 20 mM KCl, and EDTA-free protease inhibitor cocktail (Roche) at the ratio of 1 tablet per 100 ml buffer. After

centrifugation (45Ti rotor, Optima L90K, Beckman) at 125,000×g for 30 min at 4 °C, the membranes were further prepared with two washes using a high-osmotic buffer containing 10 mM HEPES, pH 7.5, 10 mM $MgCl_2$, 20 mM KCl, and 1 M NaCl, and one more wash with the hypotonic buffer to remove the high concentration of NaCl. The purified membranes were then resuspended in the hypotonic buffer supplemented with 20% glycerol, flash-frozen in liquid nitrogen, and stored at −80 °C until use.

Purified membranes were thawed on ice in the presence of 100 μM aprepitant, 2 mg ml$^{-1}$ iodoacetamide and EDTA-free protease inhibitor cocktail (Roche), and incubated at 4 °C for 1 h. The membranes were then solubilized in 50 mM HEPES pH 7.5, 10% glycerol, 500 mM NaCl, 0.5% (w/v) (DDM, Affymetrix), 0.1% (w/v) CHS (Sigma), and 50 μM aprepitant. After incubation at 4 °C for 3 h, the supernatant was isolated by centrifugation (70Ti rotor, Optima L90K, Beckman) at 125,000×g for 30 min, supplemented with imidazole to a final concentration of 10 mM, and incubated with TALON IMAC resin (Clontech) overnight at 4 °C. The resin was washed with five column volumes of wash buffer containing 25 mM HEPES, pH 7.5, 500 mM NaCl, 10% (v/v) glycerol, 0.05% (w/v) DDM, 0.01% (w/v) CHS, 30 mM imidazole, and 50 μM aprepitant, and then incubated in a Lauryl maltose-neopentyl glycol (LMNG)-exchange buffer containing 25 mM HEPES, pH 7.5, 500 mM NaCl, 10% (v/v) glycerol, 0.5% (w/v) LMNG (Affymetrix), 0.01% (w/v) CHS, 30 mM imidazole, and 50 μM aprepitant at 4 °C for 2 h. After washing with five column volumes of wash buffer, the resin was further incubated in a DDM-exchange buffer containing 25 mM HEPES, pH 7.5, 500 mM NaCl, 10% (v/v) glycerol, 0.5% (w/v) DDM, 0.01% (w/v) CHS, 30 mM imidazole, and 50 μM aprepitant at 4 °C for 1 h. Further washes were carried out with eight column volumes of wash buffer. The NK1R sample was then eluted with five column volumes of elute buffer containing 25 mM HEPES, pH 7.5, 500 mM NaCl, 10% (v/v) glycerol, 0.05% (w/v) DDM, 0.01% (w/v) CHS, 300 mM imidazole, and 50 μM aprepitant. A PD MiniTrap G-25 column (GE Healthcare) was used to remove imidazole. The receptor was then treated overnight with His-tagged PreScission protease (custom-made) and His-tagged PNGase F (custom-made) to remove the C-terminal His-tag and deglycosylate protein. The protein was subsequently incubated with Ni-NTA resin (Qiagen) at 4 °C for 1 h to remove the cleaved His-tag, PreScission protease and PNGase F. The purified NK1R-aprepitant complex was concentrated to 40 mg ml$^{-1}$ with a 100 kDa molecular weight cut-off concentrator (Millipore). Protein purity and monodispersity were tested by sodium dodecyl sulfate polyacrylamide gel electrophoresis and analytical size-exclusion chromatography (aSEC). Typically, the protein purity was over 95% and the aSEC profile showed a single peak, an indication of receptor monodispersity.

**Lipidic cubic phase crystallization.** The NK1R–aprepitant complex was crystallized using the lipidic cubic phase (LCP) method by mixing the protein sample (~40 mg ml$^{-1}$) with lipid (monoolein and cholesterol 10:1 by mass) at weight ratio of 2:3 using a mechanical syringe mixer until a homogenous mesophase was achieved[35]. The LCP mixture was then dispensed onto glass sandwich plates (Shanghai FAstal BioTech) in 40 nl drops and overlaid with 800 nl precipitant solution using a Gryphon LCP robot (Bioray GP17, ARI). Protein reconstitution in LCP and crystallization trials were performed at room temperature (19–22 °C). Crystals appeared after 3 days and reached their full size within 2 weeks in 0.1 M MES, pH 6.0–6.6, 25–35% PEG400, 200–350 mM ammonium tartrate dibasic, and 50 μM aprepitant. Crystals were harvested directly from LCP using 50–75 μm micro-loops (M2-L19-50/75, MiTeGen) and flash frozen in liquid nitrogen.

**Data collection and structure resolution.** X-ray diffraction data were collected using a Pilatus3 6 M detector (X-ray wavelength 1.0000 Å) at the SPring-8 beamline 41XU, Hyogo, Japan. The crystals were exposed with a 10 μm × 8 μm minibeam for 0.2 s and 0.2° oscillation per frame. A raster system was used to find the best-diffracting parts of single crystals[36]. Most crystals of NK1R-E78[2.50]D-aprepitant diffracted to 3.3–3.0 Å resolution and most crystals of NK1R-E78[2.50]N-aprepitant diffracted to 3.2–2.7 Å resolution. Data from the 47 best-diffracting crystals of NK1R-E78[2.50]D-aprepitant were integrated and scaled at 3.2 Å resolution using XDS[37]. Data from the 21 best-diffracting crystals of NK1R-E78[2.50]D-aprepitant were integrated and scaled at 2.7 Å resolution. Both datasets were further truncated using UCLA anisotropy server (http://services.mbi.ucla.edu/anisoscale) to remove anisotropy issues. Initial phase information was obtained by molecular replacement using the program Phaser[38] with the receptor portion of NTSR1 (PDB accession code: 4GRV) and the mT4L in the M3 structure (PDB accession code: 4U15) as templates. Refinement was performed with REFMAC5[39] and BUSTER[40] followed by manual examination and rebuilding of the refined coordinates in the program COOT[41] using both $|2F_o| − |F_c|$ and $|F_o| − |F_c|$ maps. The Ramachandran plot analysis indicates that 100% of the residues are in favorable (NK1-E78[2.50]D-aprepitant, 94.9%; NK1-E78[2.50]N-aprepitant, 96.2%) or allowed (NK1-E78[2.50]D-aprepitant, 5.1%; NK1-E78[2.50]N-aprepitant, 3.8%) regions (no outliers). The final models of NK1R-E78[2.50]D-aprepitant and NK1R-E78[2.50]N-aprepitant include 286 residues (F25-S226 and H237-R321) of NK1R and residues N1-E10 and A17-Y117 of mT4L. The remaining N- and C-terminal residues were disordered and not refined.

**NMR spectroscopy**. NMR samples were prepared using a similar procedure as for the crystallographic studies, except that the concentration of NaCl was lowered to 300 mM and no additional aprepitant or glycerol was added during the wash and elusion steps. $^{19}F$-NMR spectra of protein samples with a final concentration of 2 mg ml$^{-1}$ were measured with a Bruker AVANCE 600 spectrometer at 25 °C, using a TCI $^{1}H/^{19}F-^{13}C-^{15}N$ triple resonance cryoprobe. The $^{19}F$ chemical shifts were calibrated relative to trifluoroacetic acid (TFA) at −75.5 ppm. 1D $^{19}F$-NMR spectra were recorded with 16k complex points, which were zero-filled to 32k points; the relaxation delay was 1 s.

2D [$^{19}F,^{19}F$]-EXSY experiments were recorded with 2k and 32 complex points in the direct and indirect dimensions, respectively. The mixing time was 150 ms. The data were zero-filled to 4k and 64 points in the direct and indirect dimensions. The line-broadening factor of the EM function was 30 Hz.

**cAMP and IP1 assays**. Flag-tagged wild-type and mutant NK1Rs were cloned into the expression vector pcDNA3.1/V5-His-TOPO (Invitrogen) and expressed in HEK293 cells (ATCC). Cells were routinely tested for mycoplasma contamination. Cells were harvested 48 h post-transfection. To measure cell-surface expression of the NK1 receptors, 10 μl cells were mixed with 15 μl Monoclonal ANTI-FLAG M2-FITC antibody (Sigma, F4049; 1:100 diluted by TBS supplemented with 4% BSA). After a 20-min reaction, the fluorescence of the bound antibody was measured by an FCM (Flow Cytometry) reader (Millipore).

cAMP accumulation was measured using a cAMP kit (Cisbio Bioassays, 62AM4PEB). The harvested cells were plated into 384-well plates (6000 cells per well) and treated with different concentrations of SP (1 pM–10 μM) diluted in Dulbecco's Modified Eagle Medium (DMEM) supplemented with 0.1% BSA at 37 °C for 30 min. Then cryptate-labeled anti-cAMP monoclonal antibody and d2-labeled cAMP in Lysis Buffer were added to the wells. After a 1-h incubation at room temperature, plates were read in an EnVision multilabel plate reader (PerkinElmer) with excitation at 320 nm and emission at 620 and 665 nm. The accumulation of cAMP was calculated according to a standard dose–response curve.

IP1 accumulation was measured using an IP1 kit (Cisbio Bioassays, 62IPAPEB). The harvested cells were plated into 384-well plates (6000 cells per well) and treated with different concentrations of SP (10 pM–100 μM) diluted in DMEM supplemented with 0.1% BSA at 37 °C for 30 min. For the aprepitant competition assays, an additional 100 nM aprepitant was added and co-incubated with SP at 37 °C for 30 min. Then cryptate-labeled anti-IP1 monoclonal antibody and d2-labeled IP1 in Lysis Buffer were added to the wells. After 1 h incubation at room temperature, plates were read in an EnVision multilabel plate reader (PerkinElmer) with excitation at 320 nm and emission at 620 nm and 665 nm. The accumulation of IP1 was calculated according to a standard dose–response curve using GraphPad Prism 5.0 (GraphPad Software). The curves were normalized to the top (100%) and bottom (0%) values of the associated NK1R curve. Using nonlinear regression (curve fit) the $EC_{50}$ and $pEC_{50} \pm$ S.E.M. were calculated.

## Data availability

Atomic coordinates and structure factors of NK1R ($E^{2.50}N$)-aprepitant and NK1R ($E^{2.50}D$)-aprepitant complex structures have been deposited in the Protein Data Bank with accession codes 6J20 and 6J21. The source data underlying Fig. 3b, c and Supplementary Figs. 2 and 3 are provided as a Source Data file. Other data are available from the corresponding authors upon reasonable request.

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

## Acknowledgments

This work was supported by the National Key R&D Program of China 2018YFA0507000 (B.W. and Q.Z.), the Key Research Program of Frontier Sciences, Chinese Academy of Sciences Grant numbers QYZDB-SSW-SMC054 (Q.Z.) and QYZDB-SSW-SMC024 (B. W.), and the National Science Foundation of China grants 81525024 (Q.Z.), 31825010 (B.W.), and 31670733 (D.L.). The synchrotron radiation experiments were performed at the BL41XU of SPring-8 with approval of the Japan Synchrotron Radiation Research Institute (Proposal no. 2015B2026, 2015B2027, 2016A2517, 2016A2518, 2016B2517, and 2016B2518). We thank the beamline staff members K. Hasegawa, H. Okumura, N. Mizuno, T. Kawamura, and H. Murakami of the BL41XU for help on X-ray data collection.

## Author contributions

S.C. optimized the construct, purified the NK1 receptors for crystallization, performed crystallization trials, optimized crystallization conditions, and performed the signaling assays. M.L. helped with the construct and crystal opimization. D.L. designed the NMR part of the project. D.L. and L.Y. recorded the NMR spectra. D.L., L.Y., and K.W. analysized the NMR data and wrote the NMR part of the paper. C.Y. and L.M. expressed the NK1 receptors. H.Z. collected X-ray diffraction data. Q.L. assisted in analyzing the NK1R compound. T.F. helped with functional assays. M.-W.W. helped with ligand slection and analysis. T.S. helped to anlyze the functional assay data. R.C.S. oversaw structure analysis/interpretation and jonitly with K.W. helped with the manuscript preparation. B.W. and Q.Z. initiated the project, planned and analyzed experiments, supervised the research, and wrote the manuscript with input from all the authors.

## Additional information

**Competing interests:** The authors declare no competing interests.

