## [Peer Review File · Nature Communications]

Reviewers' comments:

Reviewer #1 (Remarks to the Author):

In their paper „Human substance P receptor 1 binding mode of the antagonist drug aprepitant revealed by NMR and crystallography“, Chen et al, present crystal structures of neurokinin 1 receptor variants bound to an antagonist, where in addition to a few stabilizing mutations, position 2.50 was mutated from Glu to Asp or Asn, respectively. The authors detect slight conformational changes and changes in the side chain orientation of residues that are located in the allosteric hydrogen bond network connecting helices 2, 3, 6 and 7. The functional relevance of the interaction between residues 2.50 and 7.49 is verified by cAMP and IP1 assays, showing that a disruption of the hydrogen network leads to reduced cellular signaling upon stimulation by SP. These insights are complemented by antagonist ligand 19F NMR experiments that demonstrate that binding to the receptor leads to the population of multiple conformational states in slow exchange, corroborating high affinity binding properties of the antagonist aprepitant. The authors identify changes in the ligand 19F signals upon mutation of the receptor at position 2.50 and conclude that this positions most likely has an impact on the ligand binding mode and affinity.

This manuscript is well written and provides novel structural information on NK1 receptor. To my knowledge, these are the first structures of NK1R so far. The combination between crystallography of two different variants and NMR together with signaling assays provide solid clues that the 2.50 position is an essential key residue for allosteric activation of a GPCR.

Nonetheless, I have a few comments and suggestions:

- 1.) I assume that the wt receptor could not be crystallized, or at least NK1R with a Glu at position 78? This would provide a good reference for the two presented NK1R variants.
- 2.) The construct used for crystallization contains four stabilizing mutations, i.e. E78D and three others. In the E78N variant, the authors mention that Q165A was reinstated. What about the other two mutations (Y121W and T222R)? This is confusing as it implies that in the latter construct, only E78D and Q165A are present.
- 3.) What was the rationale for identifying these stabilizing mutations? I could not find any comment or literature reference on this in the manuscript.
- 4.) The comparison between NK1R and NTR1 reveals major structural differences. Is the relative position of the T4L fusion protein the same in both constructs, i.e. could the T4L fusion have an impact on the conformation of helices 6 and 7? Another NTR1 structure without fusion protein and in the agonist-bound state solved by the Plueckthun lab shows that helices 6 and 7 are in a more compact arrangement, probably more similar to the conformation of NK1R presented here. Maybe it is worth discussing these discrepancies.
- 5.) The EXSY experiments with the E78N variant clearly show an exchange cross-peak between the micelle-bound and the receptor-bound ligand (P2a,b). This should be mentioned in the main text as it would demonstrate that this particular receptor variant shows weaker ligand binding.

Reviewer #2 (Remarks to the Author):

Chen et al report the crystal structure of NK1 receptor bound to antagonist aprepitant. They provide a detailed description of the aprepitant-binding site and they combine crystallography and liquid NMR to analyse the role of E78.50 and E78.50 mutants N and D (TM2) on aprepitant-bound NK1 conformation. Overall, this article is very well written and present important results about the role of E78 and surrounding residues in stabilising aprepitant binding mode but also in modulating the receptor signalling.

X-ray structures of thermostabilised mutant that included E78N/D2.50 were successfully obtained. It would be nice to present a difference map for aprepitant in both structures. The authors discuss

the interaction of mutant N and D at position E782.50 with position N7.49 (TM7) than is well known to play an important role in class A GPCRs activation mechanism. They also report a weak interaction of E78N/D2.50 with S1193.39, suggesting that the role of S1193.39 may be less important for stabilising the inactive conformation of NK1 receptor. Can the author extrapolate only based on presented results? Salt or water molecules were previously reported to link position 2.50 with 3.39. Is there any Na⁺ or water molecule at the E78N/D2.50 mutants. Additional figure panel presenting the molecular interaction of E78N/D2.50 with surrounding residues might be useful.

From NMR experiment and data analysis the authors suggest that the ligand-binding site is critically dependent on the residue in position 2.50 and they compare this data with functional experiment obtained with SP molecule. It is not clear how the E2.50 is critically important. The sentence p9, "these data suggest that the ligand binding is critically dependent on the residue in position 2.50" should be clarified for better understanding. Does E78 stabilise the inactive conformation? It is difficult to correlate the NMR data performed with antagonist with functional data recorded with agonist. Additionally, "The absence of cross peaks between P1 or P2 and M indicates that the exchange between free and receptor-bound aprepitant is too slow to be seen...". It looks like from fig4. m, that there is a cross peak between M and P2?. Could this suggest that the E78N2.50 destabilised the most favourable conformation of aprepitant bound to NK1? Without performing more experiment, I would suggest the authors to clarify this point in order to make this research article easier to access for the reader.

Finally, In the introduction, p4-line 5, "However, when E2.50 was mutated to aspartate, NK1R downstream signalling was restricted to Gq only", according Ref 17, there still some signalling for some mutations, only E78A is flat. Can the authors make the sentence consistent with literature as well as with the data presented here?

Reviewer #3 (Remarks to the Author):

Chen et al report two structures of aprepitant bound to two different mutant versions of the NK1R. This study is also supplemented with NMR and mutagenesis data. The solved structures are high resolution for GPCRs and the NMR data is potentially interesting. However, the study as a whole feels incomplete. The manuscript itself is not well structured and in places it is difficult to understand the experimental rationale and the key conclusions the authors are trying to convey.

Both x-ray structures are of mutated receptors, however there is no pharmacology included in the study to determine that the mutated constructs do not alter the affinity of aprepitant. Without this, it is difficult to determine how relevant the structures are for understanding the structure of NK1R and the mode of antagonist binding. The structures both contain mutations to residue E2.50 (one to D, one to Q). There is extensive discussion (some of which did not make sense to me) within this manuscript that describes E2.50 as essential for binding and function of NK1R. In addition, the authors report that the NMR data supports that mutations within this position alter the binding mode of aprepitant. If this is the case, then without a structure where the native E is present at 2.50, it is difficult to understand the relevance of these structures or their interpretation. At a minimum, I would expect the authors to generate a model with the native residue to compare the packing networks observed in the structures to the native receptor residue.

It is also difficult to understand the relevance of the antagonist bound structures to the discussion with respect to receptor activation. The authors have performed some mutagenesis on the 2.50-7.49 network using the agonist substance P that reveal the network is important for agonist function, however this was already known from the literature. It is difficult from what is written to understand the key conclusions the authors are drawing from their data and how this links to the conformation of the mutated networks observed in their solved structures or the NMR data given their structures are antagonist bound. It would be useful to know if the introduced mutations at position 2.50 alter the ground state of the receptor, ie, do they confer constitutive activity? The

data shown in Figure 3 is normalised, but there is no indication of how this normalisation was performed to assess if there are any differences in basal activity. It would also be useful to know if aprepitant affinity and activity is altered in these mutants (ie, do the mutants alter the affinity or kinetics of aprepitant binding and is the ligand still an antagonist at these mutants (as opposed to an agonist or inverse agonist). In addition, the structures are not valid comparisons to each other for comparing the role of E2.50 receptor and differences in the networks formed between the two structures when bound to aprepitant. The E2.50D construct contains the mutation Q4.60A (this is labelled as Q4.61 in the text, but is actually 4.60), whereas E2.50Q does not. Q4.60 is located within the aprepitant binding site in the E2.50Q structure where it forms multiple hydrogen bonds with the ligand that cannot be formed in the E2.50D structure. Only interactions are shown in the manuscript with aprepitant bound for the E2.50Q structure. With the exception of Q4.60A, are these all exactly the same in the E2.50D structure?

Within this study, there is no mutational data to confirm the predicted interactions of aprepitant observed in the structures are important for ligand affinity (albeit some interactions are supported in the NMR data). For example, the authors state that E193 and W184 greatly contribute to the binding of aprepitant. While its true in their structures that these residues form hydrogen bonds, mutations should be performed before concluding that these H-bonds greatly contribute to affinity as the trizoline group also forms other interactions with other side chains.

Mutational data,

- a) the signalling assays in figure 3 are listed as n=3 (each performed in duplicate), yet in the tables, the EC50 and Emax values are listed as n=2 performed in triplicate). Which is correct?
- b) While the functional data looks convincing. The cell surface expression data is only n=2 and is associated with a very high error (which suggests this assay has not been sufficiently optimised). This makes it impossible to determine if any of the mutations change expression levels relative to the WT. This data is crucial to interpret the effects of the mutations on signalling as efficacy is influenced by the intrinsic efficacy of the ligand for the receptor and also the expression level of the receptor. Therefore, without being confident that expression levels are not altered, the current (extensive) interpretation of the mutagenesis data on cAMP and IP1 signalling may be incorrect.
- c) Data that is n=2 cannot be reported with an S.E.M. This is SD.
- d) Page 8 in the text reports E78N/N301E displayed similar levels of cAMP and IP as the WT. If this is referring to the Emax, this statement is incorrect as cAMP accumulation is reduced by approx. 50% and IP1 by approx. 20%.

In the NMR experiments, two different substates of bound aprepitant were identified. How do these two different states relate to the binding pose identified in the structures? Some additional experiments to link the NMR data to what is observed in the static structures would strengthen the paper.

The authors discuss on page 10 residues that are presumed to provide selectivity between NK1R, NK2R and NK3R. Experimental validation is required to confirm (1) these residues are key for high affinity binding of aprepitant and (2) these are the key residues that provide selectivity. In addition, the statement that the "NK1R complex will greatly facilitate selective drugs" needs to be expanded to describe how.

Throughout the paper there are multiple grammatical errors, some repetition and some statements that need to be altered, as their meaning is ambiguous. For example

Page 8 – 1st line. "proper hydrogen interactions between helices II and VII"

Page 9: line 3. "the two peaks P2a and P2b were merged accidentally into a single peak". What does accidentally mean?

Page 10-11 – “dynamic perspective from NMR data demonstrates that aprepitant displays slightly different binding modes between the two NK1R structures, suggesting ligand efficacy may also be affected”. This statement makes no sense. The NMR and structural data use aprepitant. This is an antagonist. By definition it has no efficacy.

The final paragraph suggests the importance of position 2.50 on activation is consistent with conformational changes observed in the structures and deepen understanding of drug action and side effects associated with aprepitant administration. I’m not sure what conformational changes are consistent with activation when these are antagonist bound structures. I also fail to see how this study can inform on side effects of the antagonist.

Methods:

- a) What is the improvement in yield and thermostability of the mutant construct?
- b) Information on curve fitting and analysis for functional data needs to be included.

Figure 2: It will be useful to include a figure of the whole network here.

Responses to the reviewers' comments

1. *Reviewer #1 wondered if the NK1R with a Glu at position 78 could be crystalized, and whether the mutations Y121W and T222R existed in the E78N variant used for crystallization.*

— We thank the reviewer for the comment. We did set up crystallization trials with the NK1R protein containing E78^{2.50}. It could be crystallized, but the crystals diffracted poorly. To improve crystal quality, the mutations E78^{2.50}D and E78^{2.50}N were introduced. The mutations Y121^{3.41}W and T222^{5.64}R were included in all the constructs that were crystallized.

To make the above points clearer in the text, the following sentences were added in paragraph 1, page 5: “To improve receptor stability and facilitate crystallization, three mutations, Y121^{3.41}W, Q165^{4.60}A and T222^{5.64}R, were introduced and 10 residues (residues 227-236 of NK1R) of the third intracellular loop (ICL3) were replaced with a modified T4 lysozyme (mini-T4L). The optimized NK1R protein was co-crystallized with aprepitant, but the crystals diffracted to only about 6 Å. The resolution was improved to 3.2 Å by replacing residue E78^{2.50} with aspartic acid. To further improve the crystal quality, the residue at the 2.50 position was mutated to asparagine and Q165^{4.60}A was reinstated, whereas Y121^{3.41}W and T222^{5.64}R were maintained. The resulting NK1R-aprepitant complex structure was determined at 2.7 Å resolution”.

2. *Reviewer #1 asked about the rationale for identifying these stabilizing mutations.*

— The references for identifying the stabilizing mutations Q165^{4.60}A and T222^{5.64}R (Dodevski and Plückthun, 2007) and Y121^{3.41}W (Roth *et al.*, 2011) have now been cited in the Methods section. In brief, the Q165^{4.60} and T222^{5.64} mutations were discovered using a direct evolution method by Dodevski and Plückthun in an *E.coli* expression system. They found that a NK1R construct with ~10 mutations could significantly improve the receptor expression level and stability in *E.coli*. These mutants were expressed and purified one by one in our insect cell expression system, and the two mutants, Q165^{4.60}A and T222^{5.64}R, displayed higher protein yield and stability. The Y121^{3.41}W mutation was discovered by Roth *et al.* who found that mutating the residue at the 3.41 position to tryptophan in GPCR greatly improved the receptor stability. These 3 mutations were combined in the NK1R construct and initial crystallization conditions were obtained.

3. *Reviewer #1 questioned if the relative position of the T4L fusion protein was the same in the crystallization constructs of NK1R and NTR1, and whether the T4L fusion could have an impact on the conformation of helices 6 and 7. It was also pointed out that another NTR1 structure without fusion protein and in the agonist-bound state showed that helices 6 and 7 were in a more compact arrangement and probably more similar to the conformation of NK1R. The reviewer advised us to discuss these discrepancies.*

— We hope this explanation will clarify our position for the reviewers. First, the fusion protein technique we describe has been used to solve GPCR crystal structures for over 10 years and the majority of the GPCR structures have been determined by using this technique. Also, both active and

inactive state GPCR structures have been solved by using fusion proteins (e.g. β_2 AR), indicating that the fusion partners do not interfere with the conformational change of the receptors. In addition, it has been proven that the fusion partner did not affect the outward movement of receptors during activation (Rosenbaum *et al.*, 2007). Therefore, we conclude that neither the fusion protein nor fusion positions were likely to affect the conformation of helices V, VI or VII. In addition, in the crystal structures of NK1R-T4L and NTSR-T4L, there were several linker residues between the receptor and T4L forming flexible loop structures, which may cushion the effect of the fusion partner packing and minimize its influences on helical bundle conformation.

As the reviewer mentioned, the intracellular side of helix VI and VII showed differences in the NTSR structures with or without the T4L fusion. However, we want to emphasize that in the NTSR structure without T4L fusion, the receptor contained 11 mutations and these mutations were introduced to facilitate the binding of endogenous agonist. Several of these mutations, such as I253A, H305R and S362A, are located in the kink region of helices V, VI and VII. These mutations were intended to stabilize the receptor in an active conformation, which may be an alternative explanation for such structural differences. Another example supporting that the lack of effect of the fusion partners on receptor conformation is the fact that the inactive structures of A_{2A} AR with different fusion partners T4L (PDB ID: 3MEL) and Bril (PDB ID: 4EIY), or without any fusion (PDB ID: 3PWH), show minimum structural differences in helices VI and VII.

We hope this has clarified what we've stated in the text: "Since these structural differences were not likely induced by the fusion partner differences between the two structures, our data suggest that the NK1R-aprepitant structure is in an inactive state." (paragraph 2, page 5)

4. *Reviewer #1 suggested that the EXSY experiments with the E78N variant clearly showed an exchange cross-peak between the micelle-bound and the receptor-bound ligand (P2a,b), demonstrating that this particular receptor variant shows weaker ligand binding.*

— We thank the reviewer for this comment. However, we must emphasize that there is no exchange cross-peak between the micelle-bound and receptor-bound ligand. Here, the resonances P1 and P2 cannot be interpreted individually. Consider, if there was an exchange between the protein-bound and micelle-bound aprepitant, this exchange would involve both trifluoromethyl groups. The absence of a cross peak between the signals P1 and M, therefore, clearly shows that there is no detectable exchange (the deconvolution of the spectra in the Supplementary Fig. 4 shows that there is overlap between the signal M and one component of the signal P2, and only this component shows a cross peak with the remainder of the signal P2).

5. *Reviewer #2 advised us to present a difference map for aprepitant in both NK1R structures.*

— The advice is well taken. The electron densities of aprepitant in the NK1R structures have been included in Supplementary Fig. 1a,b.

6. *Reviewer #2 wondered if there was any Na^+ or water molecule around the E78N/D^{2.50} mutants, and suggested that we should include an additional figure panel presenting the molecular interaction of E78N/D^{2.50} with surrounding residues.*

— We didn't observe any obvious densities around the 2.50 or 7.49 residue during structure refinement. Of course, at current resolution (2.7-3.2 Å), it is very challenging to confirm any salt or water molecule in the structure. To better illustrate the local environment of the 2.50 residue, we have followed the reviewer's suggestion and added a supplementary figure showing the electron densities in this region (Supplementary Fig. 1c,d).

7. *Reviewer #2 wondered if the NMR data performed with antagonist correlated with the functional data recorded with agonist.*

— Since it is the antagonist function of aprepitant that is of significant pharmacological interest, the NMR part of this project relies on the fact that the ligand contains multiple groups of fluorine atoms and, thus it would be beyond the scope of this study to include corresponding experiments with fluorine-free agonists. Here, we merely speculate that similar binding features, with two or multiple sub-states of the bound ligand, might prevail for agonists, though, for the reasons stated, we have not addressed this in the text.

8. *Reviewer #2 pointed out that there was a cross peak between M and P2, and wondered if this suggested that the E78N^{2.50} destabilized the most favourable conformation of aprepitant bound to NK1?*

— Please see item 4 above.

9. *Reviewer #2 advised us to make the statement “However, when E^{2.50} was mutated to aspartate, NK1R downstream signalling was restricted to Gq only” in the introduction, consistent with the data in Ref 17 showing that there was still some signalling for some mutations, while only E78A completely abolished the signalling.*

— We thank the reviewer for this valuable suggestion. As shown in Ref 17, the cAMP accumulation of the E78^{2.50}D mutant was significantly affected and reduced, but it was only abolished at physiological agonist concentrations, and was recovered by using a high concentration of agonist (μM level). Therefore, to be more accurate, we have now included in the Introduction section (paragraph 1, page 4) the statement “However, it has been reported that mutating the conserved E^{2.50} to aspartic acid in NK1R reduces the Gs and β-arrestin signalling with only Gq signalling was unaffected”.

10. *Reviewer #3 suggested we generate a model with the native residue at position 2.50 to compare the packing networks observed in the structures to the native receptor residue.*

— We are grateful for this excellent suggestion. A model of NK1R with glutamic acid at 2.50 position was generated. In brief, as D/E^{2.50} is part of a highly conserved network of polar residues with several structural co-localized water molecules (and perhaps a sodium ion), the “happy” water molecule locations were first analysed in NK1R structures. Monte Carlo refinement with local minimization allowing movements to backbone and side chain atoms of the structures, including the water molecule in the location described above, was highly stable with main chain and side chains geometries in close proximity to the crystallographic coordinates. The resulting structures were

mutated back to the WT receptor, substituting D^{2.50} and N^{2.50} with E^{2.50}, and subjected to 1000 Monte Carlo steps with local minimization. The refined WT structure suggests two possibilities: (1) E^{2.50} directly interacts with both N^{1.50} and N^{7.49}; or (2) E^{2.50} makes water-mediated interactions (compared to corresponding residues of D^{2.50} and N^{2.50} in the crystal structures) not only with S^{3.39} but also with N^{7.49}. However, based on the current data it is difficult to distinguish between these two states. So we didn't include this discussion in the manuscript.

11. Reviewer #3 suggested us to test if the introduced mutations at position 2.50 alter the ground state of the receptor.

— Per suggestion above, we have measured the basal activity of NK1R with either E78^{2.50}D or E78^{2.50}N mutation using the IP1 accumulation assay. The results show that these two mutants displayed similar level of IP1 production as the wild-type receptor in both apo and aprepitant-bound states, indicating that the mutants do not affect the ground state of the receptor. These data have been included as Supplementary Fig. 3 (see figure below). To make this point clear in the manuscript, the statement “Our results of cell signalling assay indicate that mutations E78^{2.50}D and E78^{2.50}N do not influence basal activity of NK1R (Supplementary Fig. 2 and Supplementary Fig. 3)” has been added in paragraph 3, page 7.

• **Supplementary Fig. 3 | IP1 accumulation of wild-type NK1R (WT) and the E78^{2.50}D and E78^{2.50}N mutants in apo state and in the presence of aprepitant (at 1 μM concentration) or SP (at 1 μM concentration).** Data shown are mean ± S.E.M. from three independent experiments performed in technical triplicate.

12. Reviewer #3 asked whether aprepitant activity or affinity was altered by the 2.50 mutations.

— To investigate possible effect of the mutations on aprepitant activity in response to this question, we measured the substance P induced-IP1 accumulation of several NK1R mutants, including the single site mutants E78^{2.50}D, N301^{7.49}Q, E78^{2.50}N and N301^{7.49}E and the double mutants E78^{2.50}D/N301^{7.49}Q and E78^{2.50}N/N301^{7.49}E. The data show that all the mutations had little effect on the antagonism activity of aprepitant on NK1R signalling, suggesting that these mutations unlikely alter the aprepitant binding affinity to the receptor. These data have now been added to Supplementary Table 3.

Additionally, according to the basal activity data mentioned above (Supplementary Fig. 3), the IP1 production of aprepitant-bound receptor was not affected by the mutations at position 2.50,

demonstrating that aprepitant still behaves as an antagonist at these mutants.

13. *Reviewer #3 raised a concern that the two NK1R structures are not valid for comparing the role of E^{2.50} and differences in the networks formed between the two structures when bound to aprepitant, as the E^{2.50}D construct contains the mutation Q^{4.60}A while the E^{2.50}N construct does not.*

— We thank the reviewer for this comment. Indeed, the mutation Q^{4.60}A was reinstated in the E^{2.50}N construct to improve crystal quality. To exclude the influence of this mutation on aprepitant binding, the aprepitant-bound protein samples with glutamine or alanine at 4.60 position were tested using the NMR assays to measure the chemical shift of the ¹⁹F atoms in aprepitant (Fig. 4). The results showed only a subtle difference between the two samples, suggesting that the 4.60 residue does not affect the binding environment of aprepitant.

14. *Reviewer #3 suggested we perform mutational studies to confirm the key interactions of aprepitant binding observed in the structures.*

— As suggested, we have studied the effects of some mutations within the ligand binding pocket on the antagonism activity of aprepitant. Using the IP1 accumulation assay, it was observed that the antagonist aprepitant greatly inhibited the signalling of the receptor. However, when a key mutation, such as I113^{3.33}M, V116^{3.36}M, Q165^{4.60}A, E193^{5.35}H, I204^{5.46}V, F264^{6.51}Y or F264^{6.51}L, was introduced, the inhibitory activity of aprepitant on receptor signalling was greatly reduced. These data support our finding that these residues, which form close contacts with NK1R in the crystal structures, greatly contribute to aprepitant binding. These data have now been added to Supplementary Table 3. And based on these data, two sentences have now been added in the manuscript “...and mutations of the key residues greatly weaken the antagonism activity of aprepitant (Supplementary Table 3).” (paragraph 2, page 6)

15. *Reviewer #3 pointed out that the signalling assays in Fig. 3 are listed as n=3 (each performed in duplicate), while in the tables, the EC₅₀ and Emax values are listed as n=2 performed in triplicate).*

— We apologize for the typos. For all the signalling assays, at least 3 independent experiments were performed in technical triplicate. The typos in the legends of Fig. 3, Supplementary Table 2 and Supplementary Table 3 have been corrected.

16. *Reviewer #3 pointed out that the cell surface expression data were measured with n=2, which makes it impossible to determine if any of the mutations change expression levels relative to the WT.*

— To address the reviewer’s concern, we performed 3 additional independent measurements of the mutants. The results show that the deviation error is reasonable, and all the mutations had little effect on cell surface expression of the receptor. The new data have been added to Supplementary Fig. 2 (see figure below).

- **Supplementary Fig. 2 | Cell-surface expression of wild-type (WT) and mutant NK1 receptors in HEK293 cells.** Data are shown as means \pm S.E.M. from three independent experiments using independently transfected cells and performed in triplicate.

17. Reviewer #3 pointed out that $n=2$ could not be reported with S.E.M.

— We appreciate the reviewer’s comment. To improve data reliability, we performed at least 3 independent experiments for all the assays.

18. Reviewer #3 pointed out that the statement “the double mutant E78N/N301E double mutant displayed similar levels of cAMP and IP as the WT” was incorrect as the cAMP accumulation is reduced by approx. 50% and IP1 by approx. 20%.

— We thank the reviewer for the careful checking. To reflect the difference between the double mutant and WT receptors, the statement has been changed to “These results indicate that hydrogen bond interactions between helices II and VII are required for receptor activation. Similarly, the other two single mutations, E78^{2.50}N and N301^{7.49}E, caused severe loss of the SP-induced G_s and G_q signalling, while the double mutant E78^{2.50}N/N301^{7.49}E restored downstream cAMP and IP accumulation levels (to 50-80%) when compared to the wild-type receptor.” (paragraph 1, page 8)

19. Reviewer #3 questioned how two different sub-states of bound aprepitant identified in the NMR experiments related to the binding pose identified in the structures.

— Ring current calculations, which are based on the crystal structures, are the most direct way to demonstrate that the ligand-binding pocket seen in the crystal structures is also functional in solution. The NMR data further show that within this binding site there are multiple sub-states of the bound ligand that exchange with rates on the millisecond timescale.

20. Reviewer #3 advised us to provide additional experimental validation data for the speculation of aprepitant selectivity of NK1R.

— We have followed the reviewer’s advice and investigated the effect of the residues, which are presumed to provide selectivity for aprepitant, on the antagonism activity of aprepitant. We did so

using the IP1 accumulation assay. The results show that the mutation F264^{6.51}Y greatly reduced the inhibitory effect of aprepitant on NK1R signalling. These data validate our assumption that “..., the residue F264^{6.51} forms a strong edge- π interaction with the bis-trifluomethyl-phenyl ring of aprepitant, and the substitution of the phenyl group with the phenolic group may disrupt the interaction and cause a spatial clash with the ligand.” (paragraph 2, page 10). In addition, the other unconserved residues such as V116^{3.36}, E193^{5.35} and I204^{5.46}, were mutated to their corresponding residues in NK2R or NK3R, and these mutants showed only minor influences on the inhibitory effect of aprepitant on IP1 accumulation. These data suggest that the residue F264^{6.51} does play a key role in the selectivity of aprepitant. The new data have been included in Supplementary Table 3, and the following sentence was added to the manuscript: “When F264^{6.51}Y was introduced, the inhibition in G_q signalling of NK1R by aprepitant was significantly decreased (Supplementary Table 3).” (paragraph 2, page 10)

21. *Reviewer #3 suggested we clarify our statement that the NK1R complex structure will help for better selectivity drugs.*

— We appreciate this suggestion and have changed the sentence “Thus, insights gained from the NK1R-aprepitant complex structures will greatly facilitate the development of more selective drug leads” to “Thus, insights gained from the NK1R-aprepitant complex structures will greatly facilitate the development of more selective drug leads by targeting the variable regions of the ligand-binding pocket and enhancing the interactions with corresponding residues.” (paragraph 2, page 10)

22. *Reviewer #3 pointed out that the statement "proper hydrogen interactions between helices II and VII" in page 8 was ambiguous.*

— This statement has been changed to “hydrogen bond interactions between helices II and VII” in paragraph 1, page 8.

23. *Reviewer #3 questioned about the meaning of the word “accidentally” in the statement “the two peaks P2a and P2b were merged accidentally into a single peak”.*

— Our conclusion that the coalescence of P2a and P2b in NK1R[N^{2.50}] is accidental comes from our observation that the signal P1 retains two components that represent two different conformational sub-states. Thus, these accidentally have the same P2 chemical shifts. So we prefer to use “accidentally” here.

24. *Reviewer #3 pointed out that it was not correct to use “efficacy” in the sentence "In addition, a dynamic perspective from NMR data demonstrates that aprepitant displays slightly different binding modes between the two NK1R structures, suggesting that ligand efficacy may also be affected".*

— To address the reviewer’s concern, we have changed the word “efficacy” to “antagonism activity” as in “In addition, a dynamic perspective from NMR data demonstrates that aprepitant displays slightly different binding modes between the two NK1R structures, suggesting that ligand antagonism activity effect may also be affected.” in paragraph 1, page 11.

25. *Reviewer #3 pointed out that the statement “These findings are consistent with the above conformational changes observed in the NK1R structures and valuable to deepen our understanding*

of the drug action as well as multiple side-effects associated with aprepitant administration” was speculative.

— We thank the reviewer for this comment. We have removed this speculation and changed the statement to “Our observations of conformational changes in the NK1R structures, which are consistent with these results, together indicate that this hydrogen-bond network may play a key role in receptor activation and are valuable for deepening our understanding of the drug’s action.” (paragraph 1, page 11)

As for the receptor activation, since the 2.50 residue is very important for receptor signalling, we speculate on the possibility that a better understanding of how the mutations of this residue affect the ligand-binding behavior reflected by an antagonist, might prevail for agonists.

26. *Reviewer #3 asked about the improvement in yield and thermostability of the mutant construct.*

— Compared to the construct without any stabilizing mutation (containing the mT4L fusion and the C-terminal truncation), the protein yield of the mutant construct used for crystallization was improved by 2-3-fold, and the protein melting temperature measured by a protein thermostability assay (Alexandrov *et al.*, 2008) was increased by ~10 degrees. We have added these data to the Methods section with the statement: “The NK1R-mT4L gene was further modified by introducing four mutations: E78^{2.50}D, Q165^{4.60}A, Y121^{3.41}W and T222^{5.64}R (construct NK1R-E78^{2.50}D) to improve the protein yield by over 2-fold and the protein melting temperature by ~10 degrees.” (paragraph 1, page 19)

27. *Reviewer #3 suggested we include information on curve fitting and analysis for functional data.*

— We followed the reviewer’s suggestion and added the following statement to the Methods: “The accumulation of IP1 was calculated according to a standard dose–response curve using GraphPad Prism 5.0 (GraphPad Software). The curves were normalized to the top (100%) and bottom (0%) values of the associated NK1R curve. Using nonlinear regression (curve fit) the EC₅₀ and pEC₅₀ ± S.E.M. were calculated.” (paragraph 1, page 24)

28. *Reviewer #3 advised us to include a figure of whole network in Fig. 2.*

— The advice is well taken. We have added a figure of whole interaction network as Supplementary Fig. 5.

REVIEWERS' COMMENTS:

Reviewer #1 (Remarks to the Author):

The authors addressed all my concerns. However, it is still requested to include references that corroborate the statement that T4L does not perturb the conformation of the GPCR, as mentioned in the rebuttal letter.

Reviewer #2 (Remarks to the Author):

The authors have addressed all my comments. The research article is suitable for publication.

Reviewer #3 (Remarks to the Author):

The manuscript has been significantly revised since the first submission, with additional experiments included in the revision, as well as changes within the text to improve the readability. However, the following minor points still need to be addressed.

The authors have now included data to demonstrate that the E2.50 mutants do not alter basal activity of the receptor when assessed in an IP1 assay. However, ref17 of the manuscript reports constitutive activity in a cAMP accumulation assay for the E2.50 mutant used in the higher resolution structure. The authors should discuss this within the manuscript.

I suggested in my previous report that a model to compare the wildtype residue (E2.50) should be included for comparison of the environment surrounding this residue with that of the crystallised mutants, given the importance of this network in receptor function. While the authors report in their rebuttal that they have completed these experiments, they were not included as the results were ambiguous. Since the original submission, a structure of the Nk1R bound to a different ligand has been published in PNAS. Here the residue at position 2.50 was not mutated. It would be useful to compare the environment of the native E2.50 in this structure with the mutated (D2.50 and N2.50) reported in this study.

I am comfortable with the argument (in response to reviewer 1) regarding the comparison of fusion partners in other structures to conclude their fusion is unlikely to influence the inactive receptor conformation. However, most structures with fusion partners are not able to be activated and therefore please reword to reflect this as the receptor here is also unlikely to be able to be activated and therefore receptor conformation is likely to be altered (it doesn't look like the mutant studies were performed with the fusion partner construct). Please also include references to support the statement (page 5) that the fusion protein is unlikely to influence the inactive conformation (based on the arguments stated in the rebuttal).

The authors have now included some mutagenesis data on residues surrounding the ligand binding pocket. However, they claim the main residue for receptor selectivity is F6.51. This mutation results in approx. 10-fold loss in the ability of the ligand to antagonise the receptor, while the ligand has approx. 1000-fold selectivity. I5.46V has a similar effect on the antagonism as F6.51Y and I would suggest this is discussed as equally important for selectivity.

While N7.49Q does result in approx. 50% decreased E_{max} of SP, there is no effect of this mutant on potency. Given expression of this mutant is also reduced by 50%, it is equally likely that the reduced E_{max} is due to less receptors at the cell surface. This discussion needs to be expanded to include this possibility (ie if the mutation disrupts ground state interactions, this could result in reduced expression at the cells surface that may be a result of less trafficking to the cell surface or

increased constitutive receptor turnover. If less receptors are available you would expect a decreased E_{max}.

Page 11 (discussion), the authors suggest that antagonism is altered in the two Nk1R structures. In fact, according to the pharmacology data E2.50D is slightly worse at antagonising than the wildtype E, and E2.50N is slightly better. However, these will not be significantly different from the antagonism seen at the wildtype. Compared to each other, there may be a 3-fold difference in the ability of the antagonist to antagonise, but with the reported error, these are unlikely to be statistically significant. As only a single concentration of aprepitant was used (a concentration that 100-fold higher than its K_d at the wildtype receptor), then some differences in the ability to antagonise the receptor may have been missed at mutant receptors. As the current data provided suggests that the antagonism is likely to be similar, this statement should be reworded as affinity is similar as the NMR data suggests the ligand can adopt more than one pose and its ability to sample these differs depending on the residue at 2.50. This should also be reflected on page 9 with discussions on the NMR data. Here, the authors state that the ligand binding is critically dependent on the residue in position 2.50. While I accept that there are two detectable sub states of the binding pose and this is altered by mutation, all the mutants at E2.50 have a similar ability (albeit maybe a 3-fold difference) to antagonise SP binding. So perhaps reword to reflect this unless statistical analysis suggests the mutants are different from wildtype and each other.

I would like to see some statistics on all the pharmacology studies and reported values (expression, potencies and E_{max} values)

Responses to the reviewers' comments

Reviewer #1

1. *The authors addressed all my concerns. However, it is still requested to include references that corroborate the statement that T4L does not perturb the conformation of the GPCR, as mentioned in the rebuttal letter.*

— The referenced paper has been added in the manuscript as the reviewer suggested.

Reviewer #3

2. *The authors have now included data to demonstrate that the E^{2.50} mutants do not alter basal activity of the receptor when assessed in an IP1 assay. However, ref17 of the manuscript reports constitutive activity in a cAMP accumulation assay for the E^{2.50} mutant used in the higher resolution structure. The authors should discuss this within the manuscript.*

— We appreciate the reviewer for this comment. As reported in ref17, the authors found that different mutations of the conserved residues, such as 2.50 and 7.49, did not show any changes of basal activity in IP1 accumulation, which is consistent with our observations. However, they did observe a slight increase in the constitutive activity in cAMP accumulation. To make this clear, a brief discussion has been added in the manuscript as “The previously reported fact that alanine substitutions of residues, such as E^{2.50} and N^{7.49}, increased the constitutive activity G_s but not G_q signalling¹⁷ is in line with our NMR results showing that only certain receptor conformation was affected by mutations of these residues. These results suggest that even though different G protein subtypes share similar structure scaffold, their activation requires different receptor conformation that is regulated by this hydrogen-bond network” (paragraph2, page11).

3. *I suggested in my previous report that a model to compare the wildtype residue (E^{2.50}) should be included for comparison of the environment surrounding this residue with that of the crystallised mutants, given the importance of this network in receptor function. While the authors report in their rebuttal that they have completed these experiments, they were not included as the results were ambiguous. Since the original submission, a structure of the Nk1R bound to a different ligand has been published in PNAS. Here the residue at position 2.50 was not mutated. It would be useful to compare the environment of the native E^{2.50} in this structure with the mutated (D^{2.50} and N^{2.50}) reported in this study.*

— Per suggestion above, we have compared our structure with the recently published NK1R structure. A statement about the difference in the environment surrounding the 2.50 residue was added in the manuscript: “Compared to another recently solved NK1R structure with glutamic acid at 2.50 position, the side chain of E^{2.50} further extends toward the N301^{7.49} and further away from S119^{3.39}, in agree with our speculation” (paragraph 2, page 7).

However, please note that, owing to resolution limitation (3.4 Å), the densities of E^{2.50} are relatively poor in the structure reported in the PNAS paper (see figure below). The specific

interaction of the residue E^{2.50} remains ambiguous. So no further discussion was included in the revised version.

• **Electron densities of residues around position 2.50 and ligand in three different NK1R structures.** a-c, Electron densities of residues around position 2.50 in the NK1R structures with N^{2.50}, D^{2.50} (the current study) and E^{2.50} (the PNAS paper). a, N^{2.50}; b, D^{2.50}; c, E^{2.50}. d-f, Electron densities of ligand in the NK1R structures with N^{2.50}, D^{2.50} (the current study) and E^{2.50} (the PNAS paper). d, N^{2.50}; e, D^{2.50}; f, E^{2.50}. Electron densities are contoured at 1.0 σ from a $|2Fo| - |Fc|$ map and coloured blue.

4. *I am comfortable with the argument (in response to reviewer 1) regarding the comparison of fusion partners in other structures to conclude their fusion is unlikely to influence the inactive receptor conformation. However, most structures with fusion partners are not able to be activated and therefore please reword to reflect this as the receptor here is also unlikely to be able to be activated and therefore receptor conformation is likely to be altered (it doesn't look like the mutant studies were performed with the fusion partner construct). Please also include references to support the statement (page 5) that the fusion protein is unlikely to influence the inactive conformation (based on the arguments stated in the rebuttal).*

— The reference has been added in the manuscript as suggested. We agree with the reviewer that most GPCR structures with fusion partner are not able to be activated. However, we believe this is not mainly due to the reason that the fused receptor could not adopt activated conformation, but due to

the fact that G protein binding site is blocked by the fusion partner and receptor at activated conformation could not be stably maintained during the purification and crystallization process. To make this clear, we have reworded the statement in paragraph 1, page 8 as “To further understand the role of the hydrogen-bond interaction between the residues at positions 2.50 and 7.49 in receptor activation, we performed cAMP and inositol phosphate (IP) accumulation assays for the NK1R mutants $E78^{2.50}D$, $E78^{2.50}N$, $N301^{7.49}Q$, $N301^{7.49}E$, $E78^{2.50}D/N301^{7.49}Q$ and $E78^{2.50}N/N301^{7.49}E$ without any fusion partner, as it would block the G protein binding and the receptor could not be activated (Fig. 3b,c, Supplementary Figure 2, Supplementary Figure 3, Supplementary Table 2 and Supplementary Table 3)”.

5. *The authors have now included some mutagenesis data on residues surrounding the ligand binding pocket. However, they claim the main residue for receptor selectivity is $F^{6.51}$. This mutation results in approx. 10-fold loss in the ability of the ligand to antagonise the receptor, while the ligand has approx. 1000-fold selectivity. $I^{5.46}V$ has a similar effect on the antagonism as $F^{6.51}Y$ and I would suggest this is discussed as equally important for selectivity.*

— The advice is well taken. The statement has been changed as “Besides $F264^{6.51}$, $E193^{5.35}$ and $I204^{5.46}$, which form a hydrogen bond or hydrophobic interactions with aprepitant, are also not conserved among the neurokinin receptors (Supplementary Fig. 4)” in the manuscript (paragraph 1, page 11).

6. *While $N^{7.49}Q$ does result in approx. 50% decreased E_{max} of SP, there is no effect of this mutant on potency. Given expression of this mutant is also reduced by 50%, it is equally likely that the reduced E_{max} is due to less receptors at the cell surface. This discussion needs to be expanded to include this possibility (ie if the mutation disrupts ground state interactions, this could result in reduced expression at the cells surface that may be a result of less trafficking to the cell surface or increased constitutive receptor turnover. If less receptors are available you would expect a decreased E_{max} .*

— We thank the reviewer for this suggestion. As the reviewer suggested, we reprocessed the data with difference expression levels, the changes of the E_{max} was still predominant. A sentence was added to the paper: “The differences in the G protein signalling could be due to either direct influences of different mutants or lower surface expression caused by the disruption of ground state interactions of these mutants, which in turn alters the observed signalling.” (paragraph 2, page 8)

7. *Page 11 (discussion), the authors suggest that antagonism is altered in the two Nk1R structures. In fact, according to the pharmacology data $E^{2.50}D$ is slightly worse at antagonising than the wildtype E, and $E^{2.50}N$ is slightly better. However, these will not be significantly different from the antagonism seen at the wildtype. Compared to each other, there may be a 3-fold difference in the ability of the antagonist to antagonise, but with the reported error, these are unlikely to be statistically significant. As only a single concentration of aprepitant was used (a concentration that 100-fold higher than its k_d at the wildtype receptor), then some differences in the ability to antagonise the receptor may have been missed at mutant receptors. As the current data provided suggests that the antagonism is likely to be similar, this statement should be reworded as affinity is similar as the NMR data suggests the ligand can adopt more than one pose and its ability to sample these differs depending*

on the residue at 2.50. This should also be reflected on page 9 with discussions on the NMR data. Here, the authors state that the ligand binding is critically dependent on the residue in position 2.50. While I accept that there are two detectable sub states of the binding pose and this is altered by mutation, all the mutants at E^{2.50} have a similar ability (albeit maybe a 3-fold difference) to antagonise SP binding. So perhaps reword to reflect this unless statistical analysis suggests the mutants are different from wildtype and each other.

- The reviewer’s suggestion is well taken. The statement paragraph 2, page 11 has been changed to “In addition, a dynamic perspective from NMR data demonstrates that aprepitant displays slightly different binding modes between the two NK1R structures, suggesting that ligand could adopt multiple poses in NK1R, which might be further regulated by this hydrogen-bond network”.

Additionally, the statement in paragraph 1, page 9 has been changed to “These data suggest that the ligand binding pose preference is critically dependent on the residue in position 2.50”.